# EMODIALOGCN: A MULTIMODAL MANDARIN DYADIC DIALOGUE DATASET OF EMOTIONS

## ABSTRACT

Face-to-face audiovisual interaction is fundamental to human communication, conveying rich and spontaneous emotional expressions. However, existing multimodal dialogue datasets suffer from irregular framing, insufficient coverage of upper-body dynamics, limited emotional diversity, small scale, and a lack of genuine spontaneity. We introduce **EmoDialogCN**, a large-scale auditory–visual–emotion multimodal dataset specifically designed to capture the richness and spontaneity of real-world face-to-face dialogues. The dataset comprises 21,880 dialogue sessions performed by 119 professional actors across 20 realistic scenarios and 18 emotion categories, totaling 400 hours of recordings—the largest and most comprehensive of its kind. A novel data collection framework minimizes equipment interference and ensures authentic multimodal signals. Actors were encouraged to improvise based on their understanding of the context, allowing spontaneous emotions to emerge naturally. EmoDialogCN achieves superior quality metrics, including natural and clear emotional expressions confirmed by subjective evaluations (average inter-rater std = 0.12), lower emotion distribution deviation (0.64 vs. 5.65), consistent subject framing (52–59% occupancy), and comprehensive coverage of facial and upper-body expressions. Models trained on this dataset generate contextually appropriate facial expressions, natural body movements, and realistic speaker–listener dynamics, underscoring the value of authentic spontaneous emotional data. The dataset is publicly available at: `https://github.com/EmoDialogCN/EmoDialog` .

## 1 INTRODUCTION

Face-to-face interaction in the real world forms the basis for human social behavior. It involves not only the exchange of linguistic information but also the transmission of subtle emotional and cognitive signals, which are essential to building interpersonal trust and strengthening social connections. Among these signals, facial expressions and gestures play a crucial role in conveying emotions and regulating turn-taking. With advances in generative neural models technologies, there is growing interest in enabling realistic face-to-face audiovisual interactions between humans and digital agents in virtual environments. Such technologies have great potential across a wide range of applications, including remote education, intelligent virtual assistants, digital human entertainment, and emotional or psychological support. In order to build such systems, high-quality multimodal audiovisual dialogue datasets involving multiple participants are essential for training but remain scarce. Some existing datasets are compiled from publicly available audiovisual chat data (Song et al., 2024) or specifically recorded by researchers using online video communication platforms (Park et al., 2024). The emergence of these datasets has played an important role in the advancement of the field and has made important contributions to technological development.

However, these datasets exhibit various issues related to facial expression features: Some lack accurate emotion annotations, others include only a limited set of emotion categories, and many are small in scale—highlighting the absence of a large-scale, well-annotated dataset with rich emotional diversity; Additionally, due to the difference in field of view (FOV) between webcams and the human eye (Egamberdiyev, 2016), data captured using webcams often suffers from lens distortion at typical shooting distances, resulting in unnatural perspective effects and facial deformation; Furthermore, studies in sociology and computational fields have shown that mediated communication differs significantly from face-to-face interaction in terms of facial expression dynamics and emotional

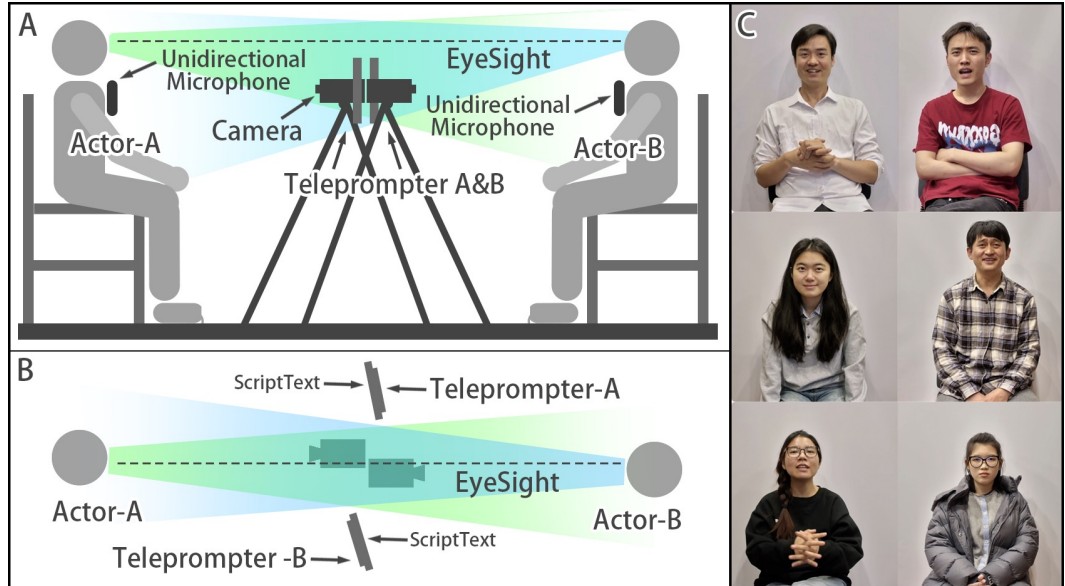

Figure 1: The layout setup of the data collection environment for the EmoDialogCN dataset (label A&B), along with some sample instances from the dataset(label C).

conveyance (Sarin et al., 2024), leads to suboptimal facial features in the collected data. These limitations can adversely affect the generative quality of downstream models.

We present the EmoDialogCN (Figure 1), a dialogue dataset designed to capture diverse and natural emotional expressions. EmoDialogCN is an auditory-visual-emotion multimodal dataset that reflects the rich emotional expressiveness of human face-to-face communication in real-world settings. It features the largest number of participants and the longest total recording duration to date. This dataset provides a solid foundation towards improving multi-turn emotional dialogue generation in multimodal settings, particularly in realistic face-to-face interaction scenarios. Its construction is guided by the following key principles.

First, the dataset adopts the emotion classification framework proposed by (Trampe et al., 2015) and employs a conversation-improvised prompt generation strategy involving both humans and AI. The resulting multi-turn dialogue segments span 20 real-life scenarios and cover 18 commonly observed emotional categories. Each dialogue session is generated from an improvised prompt and annotated with scenario labels and the overall emotional tone (e.g., angry, comforting). A rigorous validation procedure is subsequently conducted to ensure consistency between the dialogue content and the annotated atmosphere (Section 3.1).

Second, unlike conventional multi-person interaction datasets collected through real-time communication platforms, our study employs a novel recording environment specifically designed to minimize the intrusive effects of visual and audio equipment on actor performance. By using professional-grade cameras with focal lengths approximating that of the human eye, the setup provides distortion-resistant, face-to-face interactions, yielding recordings that are visually natural and perceptually consistent. This configuration not only preserves the authenticity of communication but also results in data with substantially higher quality and emotional richness (Section 3.2).

Finally, to ensure high-quality and authentic performances, we adopt two key strategies. (i) We implement real-time supervision and audio monitoring of the recording equipment to guarantee the stability and reliability of both visual and acoustic signals. (ii) Actors perform improvised dialogues based on the contextual information provided in the scripts, including topics, prompts, and the intended emotional tone. This enables them to engage in spontaneous, expressive interactions that convey genuine and contextually appropriate emotions (Section 3.3).

In summary, we present a face-to-face interaction dataset characterized by rich emotional expressiveness, along with a practical framework for data collection.

Our main contributions are summarized as follows:

- We introduce EmoDialogCN, the first high-quality, auditory-visual-emotion multimodal dataset of spontaneous and emotionally rich dialogues, designed to capture diverse emotional expressions in everyday human interactions. The dataset contains 21,800 dialogue sessions performed by 119 professional actors, covering 18 emotional categories and 20 dialogue scenarios, with a total duration of 400 hours.

- We present a novel data collection and performance authenticity–enhancing pipeline that (i) effectively eliminates distortions introduced by conventional recording setups, ensuring faithful visual and acoustic capture, and (ii) encourages actors to express their emotions authentically and naturally.

- Through observer-based subjective emotion assessment and multiple objective quantitative evaluations, EmoDialogCN demonstrates rich, authentic, and diverse emotional features.

## 2 RELATED WORK

### 2.1 PUBLIC AVAILABLE DATASETS

Training a realistic and expressive digital human audio-visual dialogue system relies heavily on high-quality datasets. Early researchers (Wang et al., 2020; Livingstone et al., 2018; Cudeiro et al., 2019; Fanelli et al., 2010) collected high-quality audio-visual data in controlled lab environments, as shown in Fig. 2, to train lip-synchronized talking face models (Prajwal et al., 2020; Fan et al., 2022; Peng et al., 2023b;a; Ng et al., 2024). While these datasets often included a wide range of emotional expressions, their limited linguistic diversity and participant variety restricted the models' generalization capabilities, leading to poor performance in real-world scenarios. To tackle this, researchers (Zhang et al., 2021; Zhu et al., 2022; Chen et al., 2025) turned to large-scale, in-the-wild datasets from the internet. These datasets offer richer linguistic diversity and a broader range of speakers, but at the cost of inconsistent data quality, often suffering from issues like lip unsynchronization and visual occlusions. Meanwhile, for emotion-centric tasks, Chinese multimodal sentiment datasets have emerged, such as CH-SIMS (Yu et al., 2020), which provide both unimodal and multimodal annotations, with a focus on modality-specific emotional perception. In contrast, our work is designed to support the development of emotionally expressive.

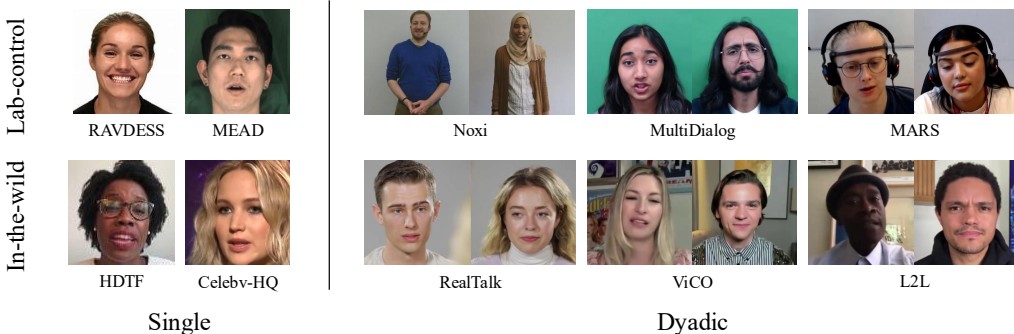

Figure 2: Publicly available datasets for single and dyadic conversation.

To achieve a more realistic interactive experience, researchers have started focusing on the digital human's listening behavior while the user is speaking. For this purpose, clips capturing the interaction between multiple subjects have been collected from movies or TV series (Poria et al., 2019; Meng et al., 2020; Wang et al., 2021). These datasets provide rich and diverse interactive data; however, the scenes in the videos are often complex, so it cannot be guaranteed that both participants' frontal facial expressions are captured. To address this issue, dyadic conversation datasets (Zhou et al., 2022; Ng et al., 2022; Geng et al., 2023; Zhu et al., 2025) specifically curated TV interviews and FaceTime sessions to ensure that both participants' facial expressions were captured simultaneously. However, since these datasets were primarily sourced from formal occasions, the facial expressions, voice tones, and behaviors captured often differ significantly from those seen in daily conversations.

Notably, works similar to ours include Noxi, MultiDialog, and MARS, which focus on dyadic interactions, capturing the nuances of human communication between two participants collected in controlled environments, ensuring data quality.

Table 1: Comparison of different datasets.

| Dataset | Type | Modality | Device | Duration (H) | Emotion annotation | Open source |
|---------|------|----------|--------|--------------|--------------------|-------------|
| RAVDESS | S | Video | WebCam | 1 | Label | ✓ |
| MEAD | S | Video | WebCam | 39 | Label | ✓ |
| HDTF | S | Video | Internet | 15 | — | ✓ |
| Celeb-HQ | S | Video | Internet | 68 | Pseudo-Label | ✓ |
| Noxi | D | Video | Kinect | 25 | Valence-Arousal | ✓ |
| ViCo | D | Video | Internet | 1 | — | ✓ |
| L2L | D | FLAME | Internet | 72 | — | ✓ |
| RealTalk | D | Video | Internet | 6 | — | ✓ |
| MultiDialog | D | Video | WebCam | 340 | Label | S only |
| DyConv | D | Video | Internet | 200 | — | ✗ |
| **Ours** | **D** | **Video** | **Varifocus Cam** | **400** | **Label** | ✓ |

EmoDialogCN excels in expression diversity, actor scale, data scale, and recording quality. S and D denote for *single* and *dyadic*, respectively.

The NoXi dataset (Cafaro et al., 2017) focused on novice–expert knowledge sharing. Experts first proposed topics they were skilled in and interested in, and novices then chose among these topics based on their own preferences. Once paired, they engaged in a natural dialogue around their shared interests. The recordings took place in two separate rooms, with Kinect2 devices used to synchronously capture multimodal signals such as video, depth maps, and skeletal data. MultiDialog (Park et al., 2024) formed its scripts from the open-domain corpus, which spans eight topics, and annotated each utterance with one of eight emotional labels. During recording sessions, two participants sat side by side and performed the scripted exchanges. Detailed acting instructions, grounded in the Facial Action Coding System (FACS) and prosodic cues, guided their performance. Example images of each emotional state were displayed on-screen to help actors accurately convey the intended affect. Whenever one participant finished speaking, they pressed a button to mark the start and end of that turn, and the listener provided natural listening reactions, enabling precise temporal segmentation and subsequent analysis. The MARS (Song et al., 2024) captured both facial video and EEG (Electroencephalography) signals. During recording, the speaker and the listener were placed in separate rooms and interacted in real time via Microsoft Teams on a screen, preventing any interference between the camera's view and the EEG equipment. The speaker led the discussion through five predefined topics, while the listener freely responded.

## 2.2 HUMAN EMOTIONAL EXPRESSION

Understanding emotional expression and social presence in both real-world and mediated interactions has been extensively explored across psychology, human-computer interaction, and affective computing. Trampe et al. (2015) conducted a large-scale ecological study on emotional experiences in daily life, identifying the frequency and distribution of 18 distinct emotions across diverse real-world contexts. Their findings serve as a foundation for emotion-aware systems by providing a sociologically grounded reference for the natural distribution of human emotional states.

In the domain of facial emotion perception, Sarin et al. (2024) used generative models to analyze behavioral differences between in-person and video-based interactions, revealing altered facial expression dynamics in virtual contexts. Kaiser et al. (2022) explored the impact of "skewed visuality" in video communication on conversational partners. The results indicated that the lack of natural eye contact disrupted the mutual intentionality of interaction, weakened emotional connection, and reduced the overall quality of dialogue. Hietanen (2018) systematically reviewed the affective impacts of gaze direction and found that direct eye contact reliably elicits heightened arousal and positive emotional responses, highlighting the crucial role of gaze in nonverbal emotional communication. Similarly, Kimmel et al. (2023) examined how facial expressions affect perceived social presence in collaborative virtual environments, confirming that subtle nonverbal cues strongly shape interpersonal connectedness even in mediated settings. The above studies suggest that communicators' facial expressions play a critical role in interactions conducted within video-based virtual environments.

## 3 METHOD

### 3.1 GENERATING CONVERSATION-IMPROVISED PROMPTS

We adopt a dialogue scenario prompting strategy to collect large-scale, high-quality dialogue data with authentic emotional expressiveness. The dialogue scenarios span 20 everyday themes, such as family, school, workplace, entertainment, and popular social topics, while the emotional atmosphere spans a wide spectrum, including relaxed, oppressive, tense, fearful, sad, and other diverse states. In this approach, trained actors first receive contextual information about the dialogue scenario, the intended emotional tone, and content-related prompts. They then engage in improvised interactions that build on this prior information, producing seamless dialogues that capture natural and emotionally rich expressions.

To facilitate this collection process, we design a practical prompt generation pipeline oriented toward improvised dialogue, ensuring that the resulting prompts are realistic and readily enactable by actors. The pipeline consists of three steps: (i) defining the dialogue scenario and intended emotional tone; (ii) generating conversation-oriented prompts with GPT-4; and (iii) iteratively monitoring and adjusting emotional coverage to achieve a balanced distribution.

In addition, we apply three design strategies to ensure the generated prompts resemble natural spoken dialogue:

1. Varied dialogue openings: diverse openers grounded in everyday situations to reduce repetitiveness and promote natural flow;

2. Theme-based prompt generation: prompts grouped by theme, offering flexible guidance for improvisation;

3. Multi-stage human review: iterative review by prompt designers, on-site supervisors, and actors to ensure authenticity and usability, while filtering out biased wording, inappropriate content, and other unsuitable information to ensure ethical safety and compliance.

Applying these strategies consistently throughout the process, we generated 21,880 dialogue sessions with substantial emotional diversity. This corpus forms the core contribution of our work: a systematic and scalable resource that provides a strong foundation for subsequent multimodal dialogue collection and modeling.

### 3.2 CONSTRUCTING THE DATA COLLECTION ENVIRONMENT

To ensure the quality of the collected data, we develop a novel data acquisition method. This approach has low interference, allowing professional actors to express natural and rich emotional characteristics more authentically and fully leverage their professional skills. The method has the following key features.

**Face-to-Face Interaction Enables More Natural Emotional Expression.** Facial expressions are integral to human communication, serving as a primary channel for the nonverbal transmission of emotional states. In face-to-face interactions, interlocutors not only perceive each other's facial expressions and vocal emotions but also observe accompanying body language, making the transmission and perception of emotions more natural and closely aligned with real-world human communication.

Therefore, to capture the most authentic facial expressions observed in everyday face-to-face communication, we position the two performing actors in a direct face-to-face setup, allowing them to see each other naturally during the interaction. Moreover, to ensure that the camera and the teleprompter do not obstruct their lines of sight, we carefully adjust their relative positions. Based on subjective evaluations by the actors, placing the camera at chest height while seated offers an optimal balance between proper framing and comfortable eye alignment. Simultaneously, the teleprompter is positioned to the side to avoid blocking the direct line of sight between the two actors. By enlarging the font size of the displayed prompts, we ensure readability without distracting from the interaction. Figure 1 shows the layout of the data collection environment: the label-A shows the side-view, the label-B shows the top-view, and the label-C shows the camera-view. By carefully configuring the positions of the cameras and teleprompters, actors can see each other face-to-face.

**High-Quality Recording Setup Addressing Visual and Acoustic Distortions.** To overcome the limitations of conventional webcams, we adopt a professional 4K high-definition camera configuration that supports vertical shooting, designed to maximize the completeness and clarity of the face and upper body. Unlike typical webcams that rely on wide focal lengths (80˜110 mm) and often introduce noticeable perspective distortion, our setup uses a focal length approximating that of the human eye, yielding a more faithful reproduction of human features and a natural visual perception. In addition, a red Safe-Frame overlay is used during recording as a real-time guide to keep actors properly positioned. After capture, the raw footage is downsampled and encoded into a 1920×1080 resolution side-by-side layout, as illustrated in Figure 3.

In parallel, to ensure clean and isolated voice capture, each actor is equipped with a dedicated unidirectional microphone, and their speech is recorded on separate audio tracks. The data collection occurs in a professionally treated sound studio, fully enclosed with acoustic panels and insulation to eliminate background noise and reverberation. In post-processing, overlapping speech segments are automatically identified and refined using audio separation and denoising techniques, further enhancing the clarity and usability of the audio data.

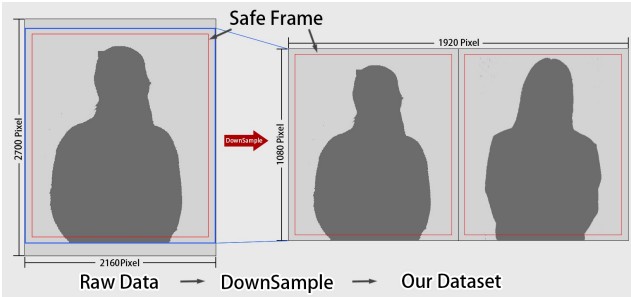

Figure 3: Framing Strategy in EmoDialogCN Dataset: Ensuring Stable Composition and High-Quality Facial & Body Capture Through Vertical 4K Recording.

### 3.3 QUALITY CONTROL AND AUTHENTIC EMOTIONAL EXPRESSION IN DATA COLLECTION.

To ensure high-quality data collection, we implemented two quality control methods during the recording process: appointing an on-site director to supervise the recording setup and monitor audio quality, and allowing actors greater creative freedom to enhance the naturalness and emotional authenticity of their performances.

**Assign an on-site director to monitor performance and data collection.** Drawing on professional television production practices, we appoint an on-site director to oversee the entire data collection process. The director provides real-time supervision of the recording setup, continuously monitoring audio quality to minimize the impact of environmental variations on the recordings. This ensures that the collected speech signals are clean and intelligible for downstream tasks such as speech recognition and emotion modeling.

**Enabling Authentic and Diverse Emotional Expression.** To ensure diversity and expressiveness, the dataset includes recordings from a wide range of performers. It includes recordings from 119 professional actors originating from 82% of the provinces and municipalities in mainland China (23 provinces and 4 municipalities). Among them, 55% are from northern regions and 45% from southern regions, ensuring broad geographic representation. This diversity naturally introduces regional accent and pronunciation variation, thereby enriching the dataset with phonetic and stylistic diversity even within the Mandarin framework.

During the performances, actors are not required to follow the prompt verbatim. Instead, they are encouraged to fully understand the scenario and atmospheric requirements of the dialogue, and then interpret their lines freely based on their understanding of the characters, the context, and their own personal style and characteristics, as long as they can authentically and naturally express emotions. This approach ensures seamless dialogues and allows emotions to emerge spontaneously, reflecting both the scene and the actors' individuality.

## 3.4 POST-PROCESSING THE COLLECTED DATA

Following data collection, we perform segmentation on the raw auditory-visual-emotion recordings. Automatic speech recognition (ASR) is applied to transcribe the audio content and generate precise timestamps for each dialogue turn. Using this alignment, the original recordings—totaling over 500 hours—are segmented into dialogue-level and turn-level clips. To ensure high data quality for downstream model training, segments with poor audio or video quality are filtered out. After this post-processing pipeline, the final dataset comprises 400 hours of high-quality, temporally aligned auditory-visual data.

# 4 RESULT

## 4.1 DATA DIVERSITY ANALYSIS

The EmoDialogCN dataset is defined by its richness in emotional, topical, and visual diversity. It covers 18 core emotions with balanced distribution, surpassing many existing corpora in both granularity and emotional coverage. The dataset spans 20 topical categories, reflecting high semantic variability and real-world relevance.

Recordings involve 119 professional actors (58 males, 61 females) aged 18–53 (average 26), ensuring diversity in voice, intonation, and speaking styles. High-resolution, wide-frame video captures allow expressive body language, encouraging actors to convey emotions through movement. To enhance visual diversity and minimize appearance-identity bias, actors vary their hairstyles and clothing across sessions.

Table 2: Detailed statistics of EmoDialogCN

| EmoDialogCN | VALUE |
|---|---|
| # actors | 119 |
| # emotions | 18 |
| # topic | 20 |
| # dialogues | 21,880 |
| # utterance | 268,404 |
| # gender (male/female) | 58 / 61 |
| avg # utterance/dialogue | 12.27 |
| avg age | 26 |
| total length (hr) | 400 |
| actor diversity (regions) | 82% provinces (55% north / 45% south) |

## 4.2 EMOTION ANALYSIS

In real life, people experience a broad spectrum of emotions on a daily basis, and the frequency of these emotions is inherently imbalanced. As highlighted in sociological research (Trampe et al., 2015), the natural distribution of emotions is heavily skewed, with neutral and mildly positive states occurring far more frequently than extreme emotions such as anger or fear. Guided by these findings, our dataset is constructed to closely mirror real-world emotional frequencies.

We further validate the emotional distribution of our dataset and conduct comparative analyses with representative multimodal dialogue datasets. As shown in Figure 4, our dataset exhibits a distribution that aligns closely with sociological statistics, while other datasets show larger deviations. To quantify this, we calculate the average absolute difference between dataset frequencies and reference human emotion frequencies across categories. Our dataset achieves a deviation score of 0.64, substantially lower than MultiDialog's 5.65, confirming its fidelity to natural human emotion patterns.

Beyond distributional validation, we conducted a subjective perception study to assess the clarity of actors' emotional expressions. 225 participants annotated 525 dialogue video clips using 18 emotion categories. These were mapped into an emotion valence vector space for statistical evaluation. Results show strong inter-rater consistency, with an overall mean standard deviation of **0.12**. For low-variance samples (std $\leq$ 0.15), the average deviation further dropped to **0.051**, indicating that emotional

expressions were both highly expressive and clearly perceived. A two-sample t-test between low- and high-variance groups confirmed a significant difference (t = -47.49, p < 0.001), reinforcing the reliability of the labels.

Together, these findings demonstrate that **EmoDialogCN** not only reflects sociologically grounded emotion distributions at the macro level but also provides micro-level evidence of expressive and perceptually clear emotional signals, ensuring strong reliability for emotion modeling and evaluation tasks.

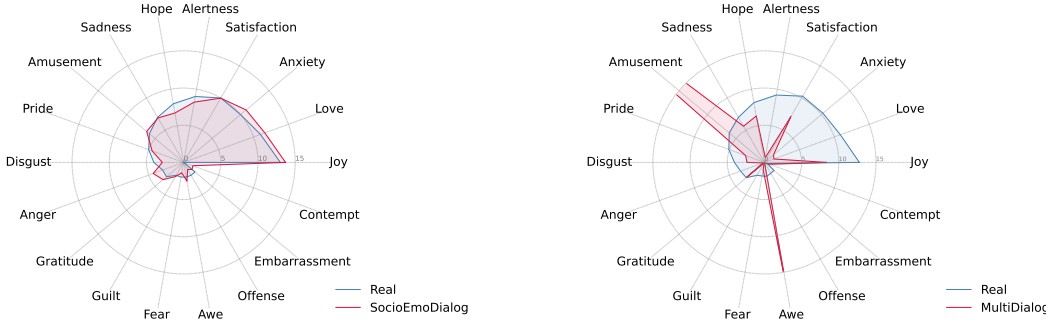

Figure 4: Comparison of emotion distributions between our dataset (left) and MultiDialog (right) (Park et al., 2024). Red bars show natural human emotion frequencies (Trampe et al., 2015), blue bars show dataset distributions. In MultiDialog, "Amusement" and "Awe" reach 40.44% and 18.17%, exceeding the 15% chart limit (red truncated).

## 4.3 BODY COMPLETENESS AND STABILITY

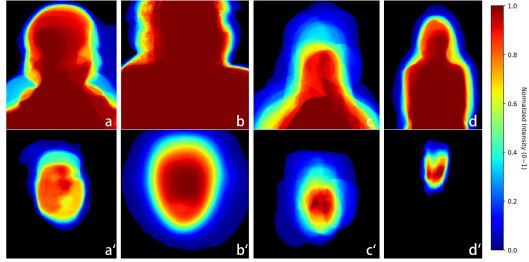

Figure 5: We sampled equal proportions of data from RealTalk, MARS, MultiDialog, and EmoDialogCN (left to right), generating body (a–d) and facial (a′, b′, c′, d′) heatmaps. The results show that EmoDialogCN achieves the greatest completeness and stability in both regions.

To improve dataset quality and reduce the amount of unusable data as well as post-processing complexity, we impose strict constraints on actor positioning during performance. Empirical measurements indicate that subjects consistently occupy approximately 52%–59% of the video frame, ensuring comprehensive coverage of both facial expressions and upper-body movements. In contrast, datasets collected using traditional webcams or short focal-length lenses often suffer from various optical and compositional issues, such as unstable subject framing, perspective distortion (e.g., exaggerated noses or foreheads, occluded ears), and the absence of upper-body visual information. We hope that this consistent framing contributes to improved data quality and facilitates broader use in downstream tasks such as facial analysis, gesture recognition, and emotion modeling.

## 4.4 DATA APPLICATION

In one of our internal experiments, we apply data from this dataset to conduct a video generation task driven by dual-dialogue audio streams. Specifically, the downstream video generation is implemented with a generative setup that combines WhisperVQ + FLUX-VAE dual-stream encoders, trained with Qwen2.5-3B, and decoded via flow-matching diffusion. As shown in Figure 6, the generated results

exhibit rich facial expression dynamics. The eyes maintain a level of focus similar to that observed in real-world face-to-face communication, and the system produces contextually appropriate facial expressions as well as natural adjustments in head and body posture in response to the audio content. Moreover, the model trained with the proposed dataset is capable of simultaneously generating videos of both speaking and listening states, thereby more closely approximating the dynamics of real-world conversational scenarios. This prototype illustrates dataset compatibility but is not the main focus of our work; systematic comparisons with other generative models remain future work.

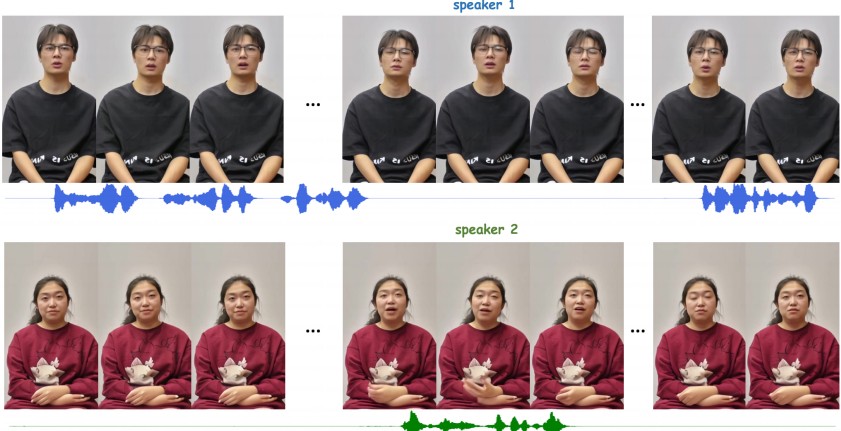

Figure 6: Data Application Effect. Generating new voice-driven video results. The figure illustrates the generated videos of two speakers in a dyadic dialogue scenario under different communicative states. It can be observed that the generated videos effectively capture both speaking and listening states, conveying a rich variety of expressions and body movements. The continuous video results can be viewed in the supplementary materials.

## 5 CONCLUSION

In this paper, we introduce EmoDialogCN, a large-scale and high-quality multimodal dataset that is designed to support the development of emotionally expressive and socially grounded auditory-visual dialogue systems. Existing datasets often suffer from limited emotional diversity, insufficient scale, and visual distortions caused by webcam-based recordings. To address these challenges, we propose a novel data collection framework and methodology, upon which we build this new dataset. Comprehensive evaluations show that EmoDialogCN outperforms existing datasets in terms of facial quality, body stability, and the overall naturalness of interaction.

Future work explores its applications in tasks such as emotion-aware response generation, emotion recognition, multimodal dialogue generation, empathetic virtual agents, and adaptive learning systems.

**Limitation.** Our dataset prioritizes high standards of emotional expressiveness and language proficiency, enabling actors to convey emotions naturally. As a result, recordings from non-native English speakers were excluded, and the dataset is currently limited to Mandarin-speaking regions. This linguistic scope restricts cross-cultural generalizability. Furthermore, we advise against direct use in clinical or neuroscience applications; explicit usage guidance will be provided in the official release.

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

## THE USE OF LLMS

In this study, LLMs were applied in two aspects: (1) Generating prompts for actors as reference. (2) Polishing the writing of the paper.

## ETHICAL STATEMENT

We take ethical concerns seriously and have designed EmoDialogCN with strong safeguards across consent, privacy, authenticity, inclusivity, and dual-use.

**Informed Consent and Privacy.** All actors participated voluntarily and signed explicit consent forms, with fair compensation agreed upon through negotiation. Each participant authorized the use of their likeness and voice for research and model training purposes. Our data practices align with international standards such as GDPR, including strict protection of personal and sensitive information. Given the risks associated with 4K video and high-fidelity audio, we adopt privacy-preserving strategies such as anonymization, modality-based tiered release, and the addition of invisible watermarking during data distribution to enhance traceability and prevent misuse. Relevant privacy-preserving pipelines will be open-sourced to ensure transparency. Access to the dataset is restricted to approved academic users through a formal application process.

**Bias and Representation.** Most existing emotion datasets are English-based and culturally Western, which limits their applicability in non-Western contexts. EmoDialogCN addresses this gap by focusing on Mandarin speakers to ensure consistency across speech, expression, and text. Nonetheless, the current version lacks multilingual participants and thus has limited cross-cultural generalizability. We clearly document this limitation and recommend further validation for cross-cultural use. Future versions will expand to Mandarin-English bilingual data and additional languages and cultures.

**Authenticity of Emotional Data.** Ensuring emotional authenticity is a key challenge. To this end, annotators, directors, and actors received training on identifying and avoiding biased or inappropriate content. Dialogue generation followed a five-stage pipeline: (1) annotators defined the emotional tone and scene context; (2) GPT-4 generated scripts; (3) human reviewers filtered out biased, offensive, or culturally inappropriate content; (4) directors and actors adapted scripts for naturalness and expressiveness; and (5) automated tools performed compliance checks. Actors were encouraged to freely interpret the prompts based on their own understanding, allowing them to improvise during performance and express authentic emotional characteristics. While the data remains semi-structured, this approach balances realism with usability. We advise against direct use in clinical or neuroscience applications, and explicit usage guidance will accompany the release.

**Dual-Use and Misuse Risks.** Recognizing dual-use concerns, the dataset is released under a CC BY-NC license, restricted to academic use via application. Enterprise-level security measures, including encryption, de-identification, digital watermarking, and activity logging, are implemented to ensure traceability and reduce misuse risks.

**Inclusivity and Cultural Sensitivity.** EmoDialogCN was intentionally developed for Mandarin speakers in Chinese contexts to address the lack of localized emotional datasets. While not intended for immediate cross-cultural use, future updates will expand to participants from Hong Kong, Taiwan, Southeast Asia, and overseas Chinese communities, with multilingual content (e.g., Cantonese, Minnan, Mandarin-English) and culturally diverse scenarios. We encourage users to apply domain adaptation before extending the dataset to global applications.

Overall, EmoDialogCN is designed to advance affective computing and emotional dialogue research while maintaining transparency, inclusivity, and strong ethical safeguards.

