# OpenReview forum: "EmoDialogCN: A Multimodal Mandarin Dyadic Dialogue Dataset of Emotions"
_ICLR.cc/2026/Conference — ICLR 2026 Conference Withdrawn Submission_

### Official Review · Reviewer_FGcg · 2025-10-16

**Soundness:** 2
**Presentation:** 3
**Contribution:** 2
**Rating:** 2
**Confidence:** 3

**Summary:**

This paper introduces EmoDialogCN, a new large-scale, multimodal dataset for Mandarin dyadic (two-person) dialogues. The primary motivation is to address the shortcomings of existing datasets, which often suffer from poor visual quality due to webcam use, limited emotional diversity, lack of spontaneity, and insufficient coverage of upper-body dynamics.

**Strengths:**

- Well-Motivated and Thoughtful Data Collection: The paper does an excellent job of identifying the key limitations of existing datasets and proposes a meticulous data collection methodology to overcome them.

- Impressive Scale: The dataset is substantial in scale, with 400 hours of recordings, 119 actors, and over 21,000 dialogue sessions.

- Clarity and Presentation: The paper is well-written, clearly structured, and easy to follow.

**Weaknesses:**

- Lack of Methodological Detail in Post-Processing and Application: The paper lacks crucial details that are necessary for reproducibility and scientific rigor. For example:
   - ASR Model: The paper states that Automatic Speech Recognition (ASR) was used to transcribe audio and generate timestamps for segmentation, but it fails to mention or cite the specific ASR model used. The performance of this model directly impacts the quality of the temporal alignments.
   - Quality Filtering: The authors mention that segments with "poor audio or video quality are filtered out," but the criteria for this filtering are not defined. This process is subjective without explicit metrics.
   - Data Application Model: The description of the video generation model is superficial, listing a series of components (WhisperVQ, FLUX-VAE, Qwen2.5-3B, flow-matching diffusion) without explaining the architecture, training methodology, or rationale for these choices.

- Superficial Experimental Evaluation: The "Data Application" section (4.4) feels more like an appendix than a core part of the results. There is no quantitative evaluation of the generated video. Metrics for lip-sync accuracy, expression appropriateness, or image quality are completely absent.

- No baselines are provided. A stronger evaluation would have involved training the same model architecture on other datasets and comparing the results to demonstrate the superiority of EmoDialogCN.

**Questions:**

Could you please specify which ASR model was used for segmentation and provide its accuracy (e.g., Character Error Rate) on a sample of your data? What were the specific, objective criteria used to filter out "poor audio or video quality" segments during post-processing?

Could you provide a more detailed diagram and description of the generative model architecture used in Section 4.4? Why was this specific combination of components chosen?

The paper explicitly mentions supplementary materials for viewing the video results. Can these be provided? The claims about the quality of the generated speaker-listener dynamics are difficult to assess from a few static images alone.

Beyond confirming the dataset's quality, have you performed any exploratory analysis that reveals unique insights? For example, given the face-to-face improvised setting, did you observe different turn-taking dynamics, gesture patterns, or listener feedback cues (e.g., nodding, smiling) compared to datasets recorded via video calls?

---

### Official Review · Reviewer_xyHx · 2025-10-31

**Soundness:** 3
**Presentation:** 3
**Contribution:** 2
**Rating:** 4
**Confidence:** 4

**Summary:**

The paper proposes EmoDialogCN, a large-scale audiovisual dataset of Mandarin dyadic dialogues, collected with an in-studio, face-to-face setup intended to minimize webcam distortions and capture upper-body dynamics. The dataset is comprised of ~400 hours, with ~21.8k dialogues, 119 actors, 20 scenarios and 18 emotions. The authors emphasize careful capture (4K vertical cameras, studio mics), semi-improvised scripts seeded by LLM prompts, segmentation with ASR, and several quality analyses (emotion distribution vs. sociological priors, inter-rater variability, body/framing heatmaps).

**Strengths:**

1. Collection framework to capture high quality dialogues. The paper presents a well-designed data collection framework that enables the recording of natural, face-to-face Mandarin dialogues with high audiovisual fidelity. The authors use 4K vertical cameras, studio microphones, and a sound-treated environment to minimize visual and acoustic distortion. The setup captures upper-body gestures, maintains consistent framing, and ensures good synchrony between audio and video

2. Multiple scenarios with varied emotional expressions. The dataset covers 20 everyday conversational scenarios designed to elicit 18 different emotions through semi-improvised performances

**Weaknesses:**

1. Unbalanced set of emotions to represent real-world emotion frequencies. The dataset is designed to mirror natural emotional frequency distributions but this leads to a large imbalance across classes. While realistic, this imbalance may limit the dataset’s usefulness for model training.
2. The dataset consists of Mandarin-speaking actors from China, which limits both cultural and linguistic diversity. Although the paper acknowledges this limitation, the absence of multilingual or cross-cultural data reduces the dataset’s generalizability.
3. Some information is missing, making parts of the paper less understandable or accurate in their descriptions (see "Questions" section e.g. Q2, Q3, Q4). For example, the paper does not specify the ASR model used for segmentation or its accuracy on Mandarin, nor does it clearly explain the construction of the “emotion valence vector space” or how inter-rater consistency was computed.

**Questions:**

1. Why did you construct your dataset to represent real-world frequencies instead of creating a balanced set, so that derived models could be more generalizable?
2. Sec. 3.4: Which ASR model did you use? How accurate is the model on Mandarin?
3. Sec. 4.2: You mention that the annotations were mapped into an “emotion valence vector space.” How exactly did you do this and why did you use only valence? Did you rely solely on facial expressions? If so, how did you account for paralinguistic information in speech?
4. Sec. 4.2: You state that “results show strong inter-rater consistency,” but you do not provide metric values.
5. Sec. 4.3: What are the “empirical measurements” you refer to in the text?
6. Sec. 4.4: This section is interesting and demonstrates the applicability of the dataset, but you do not include baselines, metrics, or user studies.
7. What instructions were given to the subjects and annotators? It would be interesting to include cross-dataset evaluations (train on EmoDialogCN, test on MultiDialog/RealTalk, and the reverse) to test generalizability.

Minor Comment

1. Dialog count appears as 21880 in abstract vs 21800 in contributions. Please correct it

**Details Of Ethics Concerns:**

The paper involves a large dataset of 4K audiovisual recordings of human subjects. The authors state that participants signed consent forms and that the dataset follows GDPR-aligned practices, with privacy-preserving measures such as anonymization, tiered release, and watermarking. However, some details remain unclear: how anonymization is implemented for facial and vocal data, and how consent revocation and data storage are managed.

---

### Official Review · Reviewer_ajbN · 2025-11-02

**Soundness:** 3
**Presentation:** 3
**Contribution:** 3
**Rating:** 4
**Confidence:** 4

**Summary:**

The paper presents EmoDialogCN, a large-scale Mandarin dyadic audio-visual-emotion dialogue dataset: 21,880 dialogue sessions with 119 professional actors across 20 scenarios and 18 emotions, totaling ~400 hours of recordings. It introduces a purpose-built face-to-face recording setup where actors were encouraged to improvise based on their understanding of the context, allowing spontaneous emotions to emerge naturally.

**Strengths:**

- Coverage and scope of the work is very resonable: 400h, 21.9k dialogues, 18 emotions, 20 scenarios, substantially larger and richer than prior dyadic dataset.
- Though improvised, however, recorded by professional speakers, with eye-contact preservation; professional cameras at human-eye focal length; and with studio acoustics
- Emotion distribution close to sociological priors (deviation 0.64 vs 5.65), inter-rater std 0.12; framing consistency (~52–59% occupancy)
- The methodological approach to dialogue generation is also reasonable -- integrating human + LLM (GPT-4) pipeline for dialogue generation.
- ASR-based turn alignment ensures that recording are properly segmented into dialogue-level and turn-level clips

**Weaknesses:**

Only a prototype application is presented; no systematic model comparisons or baseline results are provided. To demonstrate the dataset’s utility, quantitative evaluations are needed. Given the availability of both open- and closed-source multimodal models, baseline experiments could be conducted.

**Questions:**

- Is there any analysis how emotional tone reflected in the recorded speech? How did you ensure that? I understand that it has been done by professional speaker and there were supervision, however, how they have a reflection in the acoustics might be important to see, which will also ensure the quality of the dataset. Same goes for the facial expressions

---

### Note · Authors · 2025-11-12

I have read and agree with the venue's withdrawal policy on behalf of myself and my co-authors.